# Reduction of antimicrobial resistant pneumococci seven years after introduction of pneumococcal vaccine in Iceland

**Martha Á. Hjálmarsdóttir**[1,2,3]*, **Gunnsteinn Haraldsson**[1,2,3], **Sigríður Júlía Quirk**[1,2,3], **Ásgeir Haraldsson**[1,4], **Helga Erlendsdóttir**[1,2], **Karl G. Kristinsson**[1,2,3]

**1** Faculty of Medicine, University of Iceland, Reykjavík, Iceland, **2** Department of Clinical Microbiology, Landspitali University Hospital, Reykjavík, Iceland, **3** BioMedical Centre of the University of Iceland, Reykjavik, Iceland, **4** Children´s Hospital Iceland, Landspitali University Hospital, Reykjavík, Iceland

* hjalmars@hi.is

**Data Availability Statement:** All relevant data are within the manuscript and its Supporting Information files.

## Abstract

### Background

Penicillin non-susceptible (PNSP) and multi-resistant pneumococci have been prevalent in Iceland since early nineties, mainly causing problems in treatment of acute otitis media. The 10-valent protein conjugated pneumococcal vaccine (PHiD-CV) was introduced into the childhood vaccination program in 2011. The aim of the study was to investigate the changes in antimicrobial susceptibility and serotype distribution of penicillin non-susceptible pneumococci (PNSP) in Iceland 2011–2017.

### Methods and findings

All pneumococcal isolates identified at the Landspítali University Hospital in 2011–2017, excluding isolates from the nasopharynx and throat were studied. Susceptibility testing was done according to the EUCAST guidelines using disk diffusion with chloramphenicol, erythromycin, clindamycin, tetracycline, trimethoprim/sulfamethoxazole and oxacillin for PNSP screening. Penicillin and ceftriaxone minimum inhibitory concentrations (MIC) were measured for oxacillin resistant isolates using the E-test. Serotyping was done using latex agglutination and/or multiplex PCR. The total number of pneumococcal isolates that met the study criteria was 1,706, of which 516 (30.2%) were PNSP, and declining with time. PNSP isolates of PHiD-CV vaccine serotypes (VT) were 362/516 (70.2%) declining with time, 132/143 (92.3%) in 2011 and 17/54 (31.5%) in 2017. PNSP were most commonly of serotype 19F, 317/516 isolates declining with time, 124/143 in 2011 and 15/54 in 2017. Their number decreased in all age groups, but mainly in the youngest children. PNSP isolates of non PHiD-CV vaccine serotypes (NVT) were 154/516, increasing with time, 11/14, in 2011 and 37/54 in 2017. The most common emerging NVTs in 2011 and 2017 were 6C, 1/143 and 10/54 respectively.

### Conclusions

PNSP of VTs have virtually disappeared from children with pneumococcal diseases after the initiation of pneumococcal vaccination in Iceland and a clear herd effect was observed.

**Funding:** This work is an investigator-initiated study that was supported by grants from: - The Landspítali University Hospital Research Fund (MÁH) https://www.landspitali.is - GlaxoSmithKline Biologicals SA (KGK, HE, ÁH), https://www.gsk.com The funders had no role in study design, data collection and analysis, decision to publish, or preparation of the manuscript.

**Competing interests:** This work is an investigator-initiated study that was supported by grants from: - The Landspítali University Hospital Research Fund (MÁH) https://www.landspitali.is - GlaxoSmithKline Biologicals SA https://www.gsk.com The authors declare that this does not alter our adherence to PLOS ONE policies on sharing data and materials. Neither GlaxoSmithKline Biologicals SA, nor Landspítali University Hospital Research Fund had any role in study design, data collection and analysis, decision to publish, or preparation of the manuscript.

This was mainly driven by a decrease of PNSP isolates belonging to a serotype 19F multi-resistant lineage. However, emerging multi-resistant NVT isolates are of concern.

## Introduction

*Streptococcus pneumoniae* are a common cause of relatively mild localized infections such as otitis media but can also cause more severe infections like pneumonia and live threatening invasive diseases. Penicillin is the antimicrobial of choice for patients infected with susceptible pneumococci [1–4]. Non-susceptibility to penicillin and other antimicrobials is common in many countries and may necessitate higher doses of penicillin, or broad-spectrum antimicrobial agents [5–7].

The epidemiology of penicillin non-susceptible pneumococci (PNSP) in Iceland has been studied intensively from 1988–2010 with emphasis on pneumococcal serotypes and antimicrobial susceptibility [8, 9]. Two PNSP epidemics have been described in the country prior to implementation of the 10-valent protein conjugate pneumococcal vaccine (PHiD-CV) into the country's childhood vaccination program in 2011. Both were caused by known multi-resistant international pneumococcal lineages, CC90, Spain[6B]-2 CC236/270/320, single and double locus variants of Taiwan[19F]-14. The 6B lineage was first detected in Iceland in 1988, peaked in 1995 and subsequently gradually declined, simultaneously in all age groups [8]. The 19F lineage was first detected in Iceland in 1998. It became the dominating PNSP lineage in the country in 2004 and peaked in 2010, the last year prior to the pneumococcal vaccination. At that time, the proportion of PNSP of all pneumococcal isolates from patients was 37.7% [9].

Many of the serotypes commonly resistant to antimicrobials are targeted by the PHiD-CV, including 19F. Pneumococcal vaccination can therefore lead to decreased prevalence of PNSP both in invasive and non-invasive diseases [10–12]. However, vaccination has little or no effect on the prevalence of total pneumococcal carriage due to serotype replacement [13, 14]. Accordingly, replacement with emerging PNSP of non-vaccine serotypes is a cause for concern [15–17].

The aim of the study was to describe the antimicrobial susceptibility and serotype distribution of PNSP in Iceland from the beginning of vaccination until end of 2017.

## Materials and methods

### Study population

All the pneumococcal isolates in the study were cultured and identified at the Department of Clinical Microbiology, Landspítali University Hospital, Reykjavík. The department serves as the primary microbiology laboratory for the greater capital area of Reykjavík. In 2011, 202,341 individuals lived in the capital area (i.e. 63.5% of the country population of 318,452), 9.0% were children less than six years old. In 2017, 216,878 individuals lived in the area (i.e. 64.1% of the country population of 338,349), 7.7% were children less than six years old (http://www.statice.is/). Patients from other parts of the country often seek health services in the capital and were included in the study. The laboratory also serves as a reference laboratory for the whole country. In total, it is estimated that the laboratory serves 85% of the population of the country for pneumococcal cultures and identifications.

Infant vaccination with PHiD-CV (Synflorix®) was initiated in April 2011 in Iceland in a 2-plus-1 schedule, without catch-up. Over 97% of Icelandic children born in 2011 and later had received the primary vaccine doses in 2015. The study population can be considered unvaccinated prior to the study period, as only 2.3% of children were vaccinated with a protein

conjugated pneumococcal vaccine in 2010. The first year of the study, 2011, was used in this study as a baseline for the evaluation of vaccine effect. By December 2011 8.6% of children <4 years of age had received ≥2 doses of PHiD-CV [18].

## Bacterial isolates and samples

The study included all pneumococcal isolates identified at Landspítali in 2011–2017, except isolates from nasopharynx and throat. Repeat isolates of the same phenotype (same serotype and antibiogram) from the same patient within 30 days were excluded as they were considered to represent the same infection. The pneumococci were cultured and identified from routine patient specimens using conventional methods, i.e. plated on two 5% horse blood agar plates (Oxoid, Hampshare, UK), one incubated in a 5% $CO_2$ and the other one anaerobically. Identification was done using optochin test (and bile solubility if unclear). All isolates from 2016–2017 were also identified with MALDI-Tof.

The patients were divided into five age groups: 0–1, 2–6, 7–17, 18–64 and ≥65 years old. The isolates were grouped according to the sampling site as follows: middle ear (ME: swabs and pus from middle ear, or otorrhoea), lower respiratory tract (LRT: sputum, bronchiolar lavage, pleural fluid), sterile body fluids (SBF: blood, cerebrospinal fluid and joint fluid) and other sampling sites (OSS: mostly specimens from conjunctiva and sinuses).

## Antimicrobial susceptibility testing

Disk diffusion susceptibility testing was performed on all the isolates using the EUCAST guidelines [19] except for the first year when the CLSI Performance Standard for Antimicrobial Disk Susceptibility Tests was used [20]. The isolates were screened for penicillin non-susceptibility with 1 μg oxacillin discs. Oxacillin sensitive (≥20 mm zone) isolates were defined as susceptible to penicillin and other β-lactams. The minimum inhibitory concentration (MIC) for penicillin and ceftriaxone was measured for all oxacillin resistant isolates (<20 mm zone) using the E-test® (Solna, Sweden/bioMérieux, France) [21]. Non-susceptibility to penicillin was defined as penicillin MIC of >0.06 mg/L. Disk diffusion susceptibility testing was also performed for chloramphenicol, erythromycin, clindamycin, tetracycline and trimethoprim/sulfamethoxazole. All susceptibility results were interpreted according to the EUCAST criteria for clinical breakpoints [22].

## Serotyping

Oxacillin resistant isolates were routinely screened for vaccine serotypes using agglutination with Pneumotest-Latex and/or latex antisera for specified serotypes or pools (Statens Serum Institute, Copenhagen, Denmark) [23]. Isolates that did not belong to vaccine serotypes were serotyped using PCR. The PCR was done as sPCR, or mPCR including a panel of all the serotypes included in the PHiD-CV and selected serotypes previously detected in our studies— serotypes 1, 3, 4, 5, 6A, 6B, 6C, 6D, 7F, 8, 9V, 9N, 10A, 10F, 11A, 12F, 14, 15A, 15B/C, 16F, 17F, 18A/B/C/F, 19A, 19B/C, 19F, 20A/B, 21, 22F, 23A, 23B, 23F, 24F, 29, 31, 33F,33B/D, 34,35B, 35F, 35(25F), 42 (35A/C), 47A [8, 9, 24–26]. Serogroup/serotype-specific primer pairs were used with primer pairs for *cps*A for the *cps* locus and *lyt*A for autolysin as internal controls. In addition, primers for *cps*B were used to confirm the absence of capsular genes, when *cps*A was negative [24, 27–30]. Serotypes of serogroup 6 were identified using previously described PCR methods [31–33]. Vaccine serotypes (VT) were defined as the serotypes targeted by the PHiD-CV. Non-vaccine serotypes (NVTs) were defined as serotypes not targeted by the PHiD-CV. Isolates were defined as non-encapsulated *S. pneumoniae* (NESp) when they were PCR positive for *lyt*A and negative for the capsular genes *cps*A and *cps*B [24, 26].

## Statistical analysis

Statistical analyses were done using Fisher´s exact and chi-square tests. Statistical significance was set at p≤0.05.

## Ethics statement

The study was approved by The National Bioethics Committee (VSNb2013010015/03.07) and the appropriate authorities at the Landspitali University Hospital, Iceland. The samples were bacterial isolates and all patient data were fully anonymized before being analysed. The National Bioethics Committee waived the requirement for informed consent.

## Results

### Demographics

From 2011–2017, the laboratory received a total of 87,374 samples for culture from ME (4,868), LRT (13,633) and SBF (68,873). The total number of pneumococcal isolates from these samples that met the study criteria was 1,728. One patient with multiple identical isolates, within the two last years of the study, was considered an outlier and his repeated isolates excluded. A few isolates had not been stored or did not survive storage, making the number of available isolates for the study 1,706 (98.7%). The annual number of pneumococcal isolates decreased gradually with time, from 348 isolates in 2011 to 163 in 2017 (p<0.0001).

ME samples were 4,868 and their annual number decreased significantly during the study period. At the same time the proportion of samples with pneumococci decreased from 191/950 (20.1%) in 2011 to 36/504 (7.1%; p<0.0001) in 2017. Samples from LRT were 13,633 in total and their annual number increased significantly with time. At the same time the number of pneumococcal isolates remained relatively stable resulting in decreased proportion of samples with pneumococci, 85/1,676 (5.1%) in 2011 and 85/2,356 (3.6%; p = 0.03) in 2017. Samples from SBF were 68,873 and their annual number increased significantly. At the same time the proportion of samples with pneumococci decreased from 31/8,817 (0.4%) to 24/11,908 (0.2%). Information on the number of samples from OSS was not available, but the number of pneumococcal isolates detected decreased from 41 in 2011 to 18 in 2017 (Fig 1 and S1 Table).

Pneumococcal isolates from the youngest age group, 0–1 years old children were 523 (of the total 1706 isolates, 30.7%) and 422/523 (80.7%) originated from ME. In 2011 the number of isolates from this age group was 154/348 (44.3%) and significant reduction was already seen in 2012 when the number of isolates were 109/322 (33.3%; p = 0.007). Gradual decline continued through the remaining period to become 12/163 (30.7%) in 2017. Isolates from the oldest age group, ≥65 years old patients, were 393/1706 (23.0%) and 284/393 (72.3%) were from LRT. There was a non-significant decline in the numbers with time, 56 isolates in 2011 and 37 in 2017 (p = 0.08; Fig 2 and S2 Table).

### Susceptibility

The number of PNSP was 516 (of the total of 1,706 pneumococcal isolates, 30.2%). In 2011 the number of PNSP was 143/348 (41.1%) and decreased to 42/202 (20.8%; p<0.0001) in 2014, but then increased to 54/163 (33.1%) in 2017 (p = 0.009; S1 Table).

Temporal changes were observed in the penicillin MIC values of the PNSP with an increase in the proportion of isolates belonging to the intermediate susceptibility category that had relatively low MIC values, most often >0.06–0.5 mg/L, but a decrease in the proportion of isolates with higher MIC values. In 2011, 14/143 (9.8%) isolates had penicillin MIC value of >0.06–0.5 mg/L, 97/143 (67.8%) of >0.5–2.0 mg/mL and 32/143 (22.4%) of >2.0 mg/L. In 2017 the

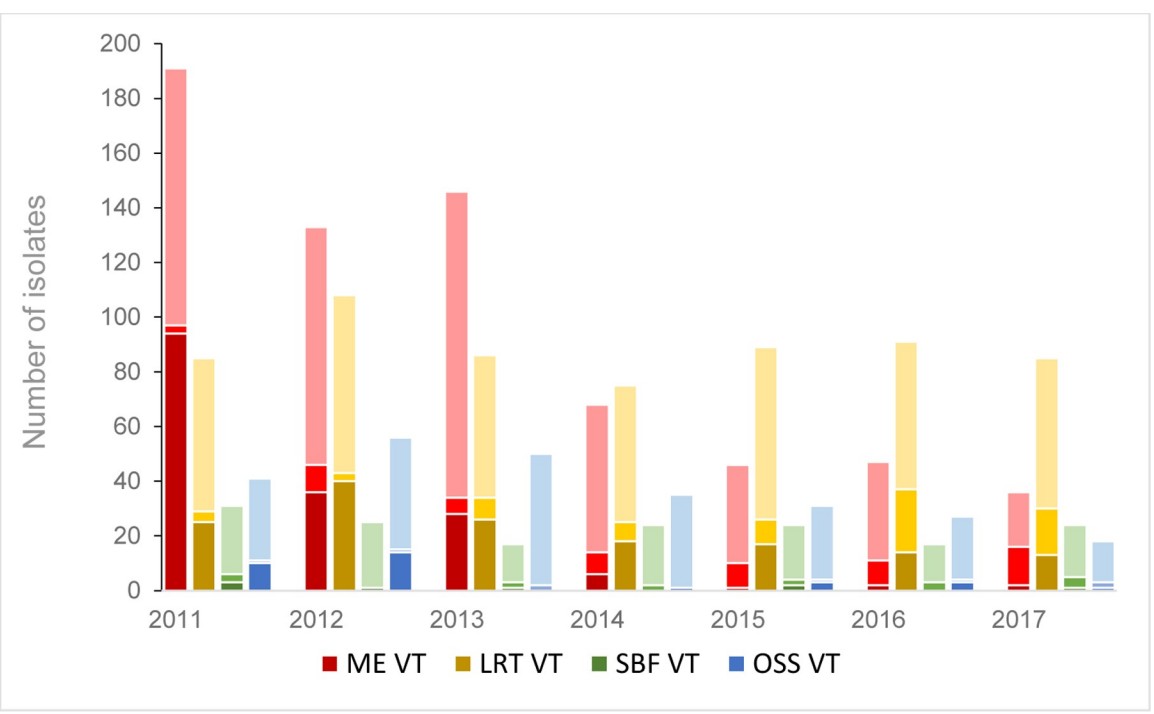

**Fig 1. Annual numbers of PNSP of vaccine serotypes, PNSP of non-vaccine serotypes and susceptible pneumococcal isolates according to sampling site.** In each cumulative column, the PNSP of VTs are shown in the darkest color at the bottom, the PNSP of NVTs in lighter color in the middle area and the PSP in the lightest color at the top of the columns. The colors differ to differentiate the sampling site presented, red for middle ear, yellow for lower respiratory tract, green for sterile body fluids and blue for other sampling sites.

numbers were 39/54 (72.2%; p<0.0001) isolates, 13/54 (24.1%; p = 0.0001) and 2/54 (3.7%; p = 0.001), respectively (Fig 3).

Of the PNSP isolates, 16/516 (3.1%) were susceptible to all routinely tested antimicrobials other than penicillin, 51/516 (9.9%) were non-susceptible to two antimicrobial classes and 449/516(87.0%) were multi-resistant (non-susceptible against three or more antimicrobial classes). Most of the PNSP isolates, 333/516 (64.5%), were non-susceptible to five antimicrobial classes. Of the multi-resistant isolates, 311/449 (69.3%) were of serotype 19F. This number decreased with time from 122/134 isolates in 2011 (91.0%) to 15/41 in 2017 (36.6%; p<0.0001; Fig 4).

PNSP isolates from the youngest age group, children 0–1 year old, were 179/516 (34.7%). The annual number and proportion decreased from 88/143 (61.5%) isolates or 1172.1 isolates/ 100,000 inhabitants in 2011 to 4/44 (9.1%) in 2015, then increased again to 12/54 (22.2%), or 194.9 isolates/100,000 inhabitants in 2017 (difference between 2011 and 2017, p<0.0001). The large majority originated from ME, 170/179 (95.0%). Two isolates were from SBF, one in 2011 and the other in 2016. PNSP isolates from the oldest age group, ≥65 years old, were 122/516 (23.6%), or 69.6/100,000 inhabitants, highest at 23/42 in 2015 (54.7%) and lowest at 12/54 (22.2%), or 35.8/100,000 inhabitants in 2017 (p = 0.001). The majority originated from LRT, 105/122 (86.1%; S2 Table).

## PNSP serotypes

PNSP isolates of VTs were 362/516 (70.2%) and of NVTs 154/516 (29.8%). The number of isolates of VTs decreased with time, 132/143 (92.3%) in 2011 and 17/54 (31.5%) in 2017

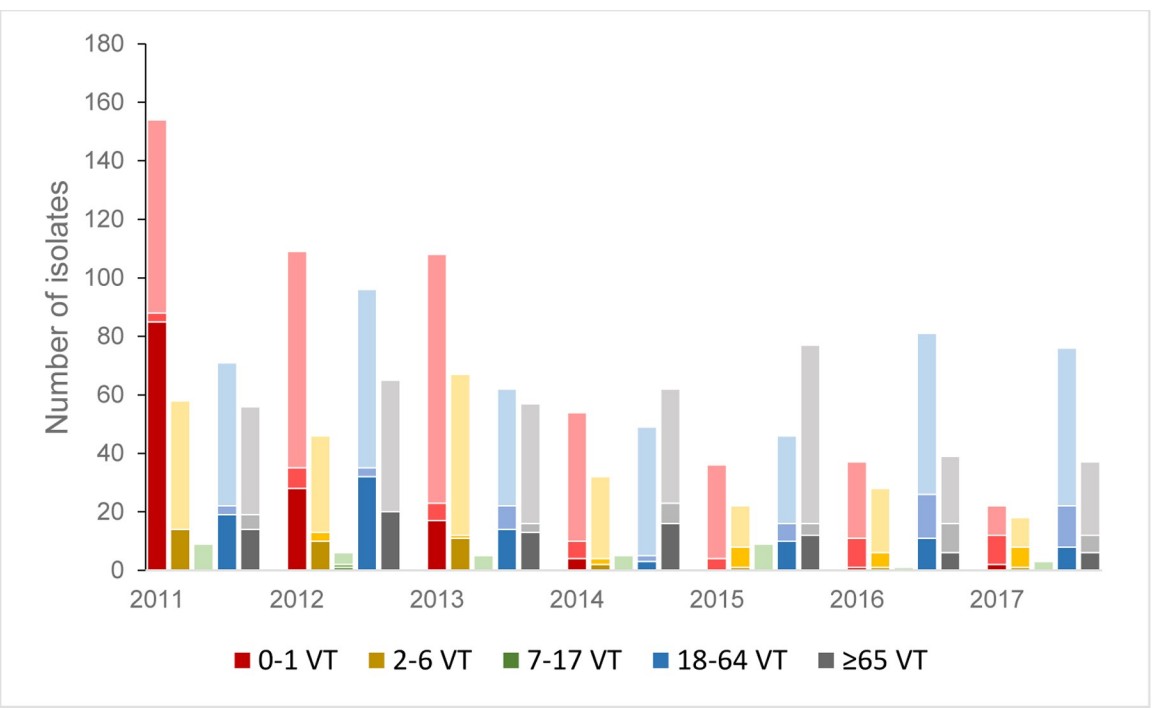

**Fig 2. Annual numbers of PNSP of vaccine serotypes (VT), PNSP of non-vaccine serotypes (NVT) and susceptible pneumococcal isolates (PSP) according to age groups.** In each cumulative column, the PNSP of VTs are shown in the darkest color at the bottom, the PNSP of NVTs in a lighter color in the middle area and the PSP in the lightest color at the top of the columns. The colors differ to differentiate the age group presented, red for 0–2, yellow for 2–6, green for 7–17, blue for 18–64 and grey for ≥65 years old.

(p<0.0001), while the number of isolates of NVTs increased from 11/143 (7.7%) in 2011 to 37/54 (68.5%) in 2017 (p<0.0001). The number of VT isolates decreased in all age groups, but first and most in the youngest children. In 2011, 85/88 (96.6%) of the PNSP in children 0–1 years old were of VTs, or 1132.1/100,000 inhabitants. In 2012 28/35 (80.0%) isolates, or 395.0/100,000 inhabitants, (p = 0.006) and in 2017, 2/12 (16.7%), or 32.5/100,000 inhabitants.

The most common serotype was 19F, 317/516 (61.4%) isolates, accounting for 87.6% of the PNSP isolates of VTs. Their number was highest in 2011, 124/143 (86.7% of PNSP), lowest in 2016, 13/55 (23.6%; p<0.0001) and were 15/54 (27.8%) in 2017. Repeated isolation of 19F with more than 30 days intervals was detected in 41 patients and the majority 28/41 (68.3%) had one repetition (Fig 5 and S3 Table).

PNSP isolates of serotype 19F from children 0–1 years old were 81/88 (92.1%) in 2011 and 1/12 (8.3%) in 2017 (p <0.0001), while in patients ≥65 years old they were 12/19 (63.2%) and 6/12 (50.0%), respectively (p = 0.71).

Other PNSP VTs identified were serotype 6B with 26/516 (5.0%) isolates, serotype 14 with 12/516 (2.3%), serotype 23F with 6/516 (1.2%) and 9V with 1/516 (0.2%) isolates.

PNSP isolates of NVTs were 154/516, 11/143 (7.7%) in 2011 and 37/163 (68.5%, p = 0.0003) in 2017. The most common serotype was 6C, 34/516 (6.6%) isolates, which increased with time. One isolate was detected in 2011, but 10/54 (18.5%) in 2017 (p<0.0001). All the isolates were multi-resistant with the following antibiogram: intermediate to penicillin with MIC of 0.094–0.125 mg/L, resistant to erythromycin, clindamycin and tetracycline, but susceptible to sulfa-trimethoprim and chloramphenicol. The second most common NVT was serotype 15A with 17/516 (3.3%) isolates, increasing with time. All the 15A isolates were multi-resistant and had similar antibiograms as the serotype 6C isolates. The third was serotype 19A with 15/516

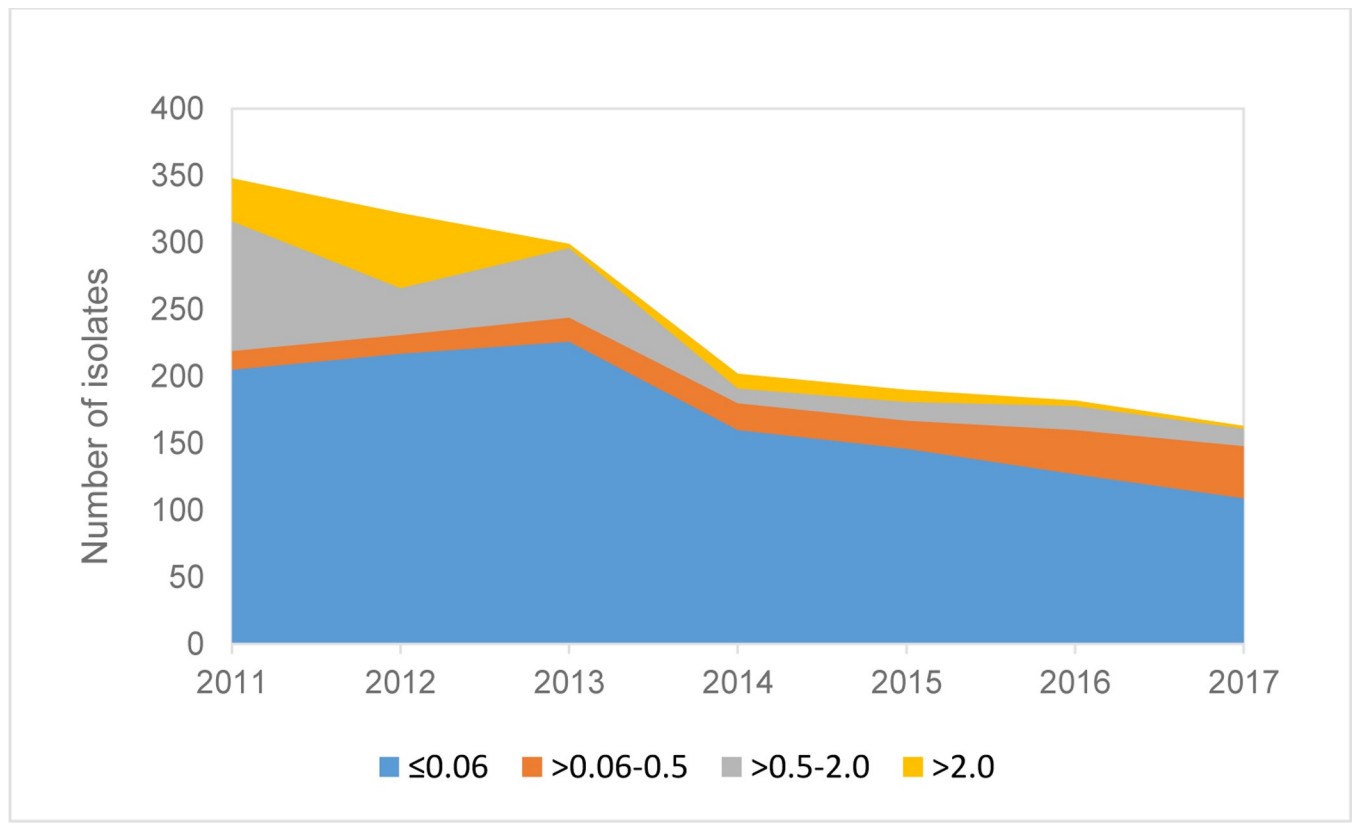

**Fig 3. The MIC values (mg/L) of all 1,706 pneumococcal isolates according to years.** Oxacillin sensitive and isolates with confirmed penicillin MIC of ≤0.06 mg/L were defined as sensitive; intermediate with relatively low MIC values, >0.06–0.5 mg/L; intermediate with relatively high MIC values, >0.5–2.0 mg/L; and resistant isolates with penicillin MICs of >2.0 mg/L.

(2.9%) isolates, 1–4 isolates per year over the study period. Non-encapsulated S. pneumoniae (NESp) isolates were 33 (6.4%), their number fluctuated with time from 2–8 isolates a year. All the NEsp isolates originated from adults except one from a 12 years old child, 30 of the isolates were from the LRT and three from conjunctiva.

## Discussion

Penicillin non-susceptible pneumococci of vaccine serotypes have virtually disappeared from children with pneumococcal diseases after the implementation of the protein conjugate pneumococcal vaccine into the childhood vaccination program in Iceland. Moreover, a clear herd effect was observed in other age groups, but least in the oldest patients.

Our group has shown in another recent study that nasopharyngeal carriage of PNSP of VTs has become rare in children since pneumococcal vaccination was initiated while pneumococcal carriage rates have not changed [24]. As carriage is the prerequisite for pneumococcal infections [34, 35] and as the nasopharynx of children is the main source of pneumococci [36, 37] these findings complement each other.

In 2011, the first year of the study, the proportion of PNSP was higher than ever before in Iceland. This peak was caused by the multi-resistant 19F lineage, more than nine out of ten PNSP isolates belonged to 19F that emerged as a dominating PNSP serotype in 2004 [9] Almost all of the remaining PNSP isolates in that year were of VTs. In the last year of the study, 2017, the number of PNSP isolates of VTs was only a fraction of the initial number with also a strong decrease in the total number of pneumococcal isolates. Isolates of serotype 19F

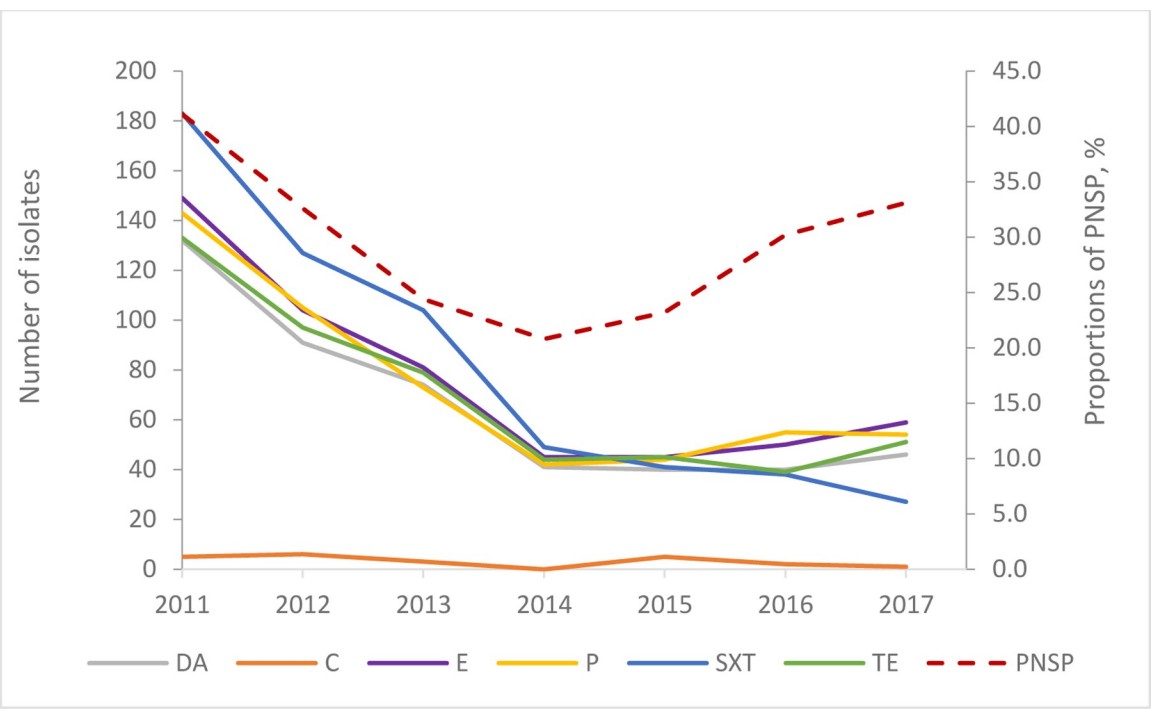

**Fig 4. Annual number and proportions of non-susceptible pneumococcal isolates to routinely tested antimicrobials.** Numbers of non-susceptible isolates to the antimicrobials are shown on the left Y-axis: clindamycin (DA) chloramphenicol (C), erythromycin (E), penicillin (P), trimethoprim/sulfamethoxazole (SXT) and tetracycline (TE).The proportions of PNSP of all pneumococci are shown in percentages on the right Y-axis (red dotted line).

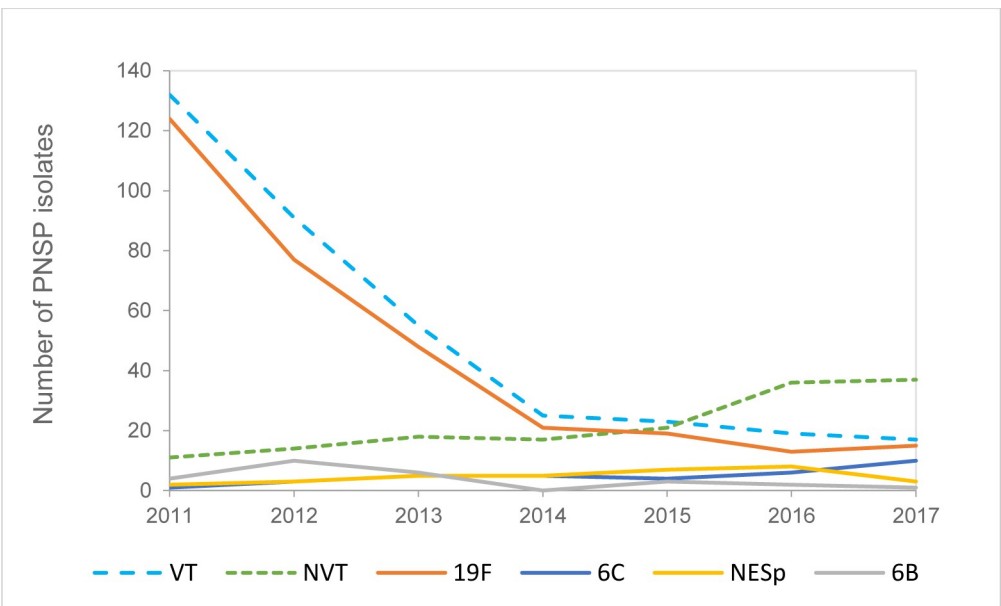

**Fig 5. Annual numbers of PNSP isolates of vaccine and non-vaccine serotypes and their four most common serotypes.** The total number of isolates of vaccine serotypes is presented by blue dotted line and of non-vaccine serotypes by green dotted line. The most common PNSP serotype was 19F, followed by 6C, non-encapsulated *Streptococcus pneumoniae* (*lyt*A positive and *csp*A and *csp*B negative) ranked third and 6B fourth.

were still the majority of PNSP of VT, although only a fraction of their number in 2011, all originating from adults except one. Isolates of other VTs in 2017 were one isolate each of serotype 6B and 14. These findings strongly demonstrate the important effect of the vaccination to reduce PNSP in children and in adults. The fact that the PNSP serotypes were initially more or less all of VTs, facilitated this dramatic change. However, serotype replacement with PNSP of NVT in the last years of the study is a cause of concern.

The decreased prevalence of PNSP was already apparent in children 0–1 year old in the second year of the study. By that time the number of PNSP isolates of VTs had decreased by more than a half from the first year. This reflects a reduction of acute otitis media following vaccination as described by Sigurdsson *et. al.* [38]. The reduction was mostly due to decreased prevalence of the multi-resistant serotype 19F of the CC236/270/320 lineage as described by Quirk *et. al.* [24, 26]. The reduction of PNSP after pneumococcal vaccine was as expected as most of the PNSP isolates in the era prior to vaccination belonged to VTs [10, 39]. In the last year of the study, PNSP of VTs were identified only in three children under the age of seven years. Isolates of serotype 19F were cultured from ME samples from two children, aged one and two years, both were outpatients and their immune and vaccine status unknown. The third isolate was of serotype 14 identified from blood culture of a fully vaccinated, chronically ill one year old child. A few incidents with PNSP of VTs could be expected, neither the efficacy of the vaccine, nor the participation and compliance to the vaccination program was absolute.

Towards the end of the study period, a significant decrease in PNSP of VTs in patients 18–64 years of age had become apparent, while the incidence of NVTs increased. This finding strongly indicates herd effect in this age group. The decrease in PNSP of VTs compared to increase of NVT in patients ≥65 years was not significant according to the number of pneumococcal isolates. However, a significantly fewer samples from this age group included isolates of VTs in the last years of the study compared to the first.

The most prominent PNSP of non-vaccine serotypes emerging after vaccine initiation were serotypes 6C and 15A. Both were usually multi-resistant, but had lower penicillin MIC values than the multi-resistant 19F lineage, explaining the reduction in penicillin MIC values with time. PNSP of serotype 19A were seen on few occasions without changes in prevalence during the study period and were not regarded problematic as has been described in other studies [40–42]. Isolates of serotype 6A were rarely seen but remained stable throughout the study period [43]. The emergence of serotypes 6C and 15A has been described in several countries and their emergence in Iceland calls for close monitoring [44–47].

Interestingly NESp were not found in children while they were among the most common pneumococci in adults. This is in contrast to findings of other investigators who have reported increase in children after initiation of vaccination [48, 49]. In an earlier study of our group a large proportion of the isolates were analysed using whole genome sequencing, confirming the identification and the absence of the capsule [26].

The effect of the financial crisis that started in Iceland in 2008 had an impact on the number of samples sent for culture. This made statistical analysis more complex, which might be considered a weakness. At the beginning of the study, physicians were still being advised to reduce diagnostic testing as much as possible. When the economic situation began to improve, around 2014, the number of samples increased to similar levels as before the crisis, then gradually increased over the last years of the study except for ME samples that continued to decline. This decline of ME samples reflects the effect of the vaccination reducing otitis media incidence in children [18, 50]. Focusing on the proportion of PNSP is not sufficient to describe the changes taking place after vaccination. The reduction of PNSP of VTs in LRT and SBF appear to be less according to the numbers of detected pneumococcal isolates rather than to the numbers of cultured samples. This affects the evaluation of the herd effect in adults, as almost all

LRT samples (95%) originated from adults. On the other hand, almost all ME samples (95%) were from children, while SBF samples reflected a broader age group, or the risk groups for invasive disease.

The continued surveillance of PNSP since they were first detected in Iceland in 1988 makes this study unique. The results demonstrate the situation for all age groups, virtually for the whole population for three decades. The vaccination has made a substantial impact on the rate of PNSP. However, emerging resistant non-vaccine serotypes are of concern and demands continued monitoring of PNSP.

## Supporting information

**S1 Table. Annual numbers and proportions of samples, total number of pneumococcal isolates, penicillin non-susceptible pneumococci (PNSP) and there of vaccine serotypes (VT) and non-vaccine serotypes (NVT), all according to sampling site.**
(DOCX)

**S2 Table. Annual numbers and proportions of samples, total number of pneumococcal isolates, penicillin non-susceptible pneumococci (PNSP) and there of vaccine serotypes (VT) and non-vaccine serotypes (NVT), all according to age groups.**
(DOCX)

**S3 Table. Annual distribution of pneumococcal serotypes and non-encapsulated *Streptococcus pneumonia* (NESp) defined as penicillin non-susceptible pneumococci (PNSP) in 2011–2017, number of isolates (n) and proportions (%).**
(DOCX)

## Acknowledgments

We thank the staff at the Department of Clinical Microbiology, Landspitali University Hospital and students at the University of Iceland involved in this project and related projects.

## Author Contributions

**Conceptualization:** Martha Á. Hjálmarsdóttir.

**Data curation:** Martha Á. Hjálmarsdóttir.

**Formal analysis:** Martha Á. Hjálmarsdóttir.

**Funding acquisition:** Martha Á. Hjálmarsdóttir, Ásgeir Haraldsson, Helga Erlendsdóttir, Karl G. Kristinsson.

**Investigation:** Martha Á. Hjálmarsdóttir, Gunnsteinn Haraldsson, Sigríður Júlía Quirk.

**Methodology:** Martha Á. Hjálmarsdóttir, Gunnsteinn Haraldsson, Sigríður Júlía Quirk.

**Project administration:** Martha Á. Hjálmarsdóttir, Gunnsteinn Haraldsson, Karl G. Kristinsson.

**Resources:** Martha Á. Hjálmarsdóttir, Ásgeir Haraldsson, Helga Erlendsdóttir, Karl G. Kristinsson.

**Supervision:** Martha Á. Hjálmarsdóttir, Karl G. Kristinsson.

**Validation:** Martha Á. Hjálmarsdóttir, Gunnsteinn Haraldsson.

**Visualization:** Martha Á. Hjálmarsdóttir, Gunnsteinn Haraldsson.

**Writing – original draft:** Martha Á. Hjálmarsdóttir, Karl G. Kristinsson.

**Writing – review & editing:** Martha Á. Hjálmarsdóttir, Gunnsteinn Haraldsson, Sigríður Júlía Quirk, Ásgeir Haraldsson, Helga Erlendsdóttir.

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
