## [Decision Letter · Decision Letter 0]

4 Dec 2019

PONE-D-19-30435

Reduction of antimicrobial resistant pneumococci seven years after introduction of pneumococcal vaccine in Iceland

PLOS ONE

Dear Professor Hjálmarsdóttir,

Thank you for submitting your manuscript to PLOS ONE. After careful consideration, we feel that it has merit but does not fully meet PLOS ONE’s publication criteria as it currently stands. Therefore, we invite you to submit a revised version of the manuscript that addresses the points raised during the review process.

We invite you to submit a revised version of the manuscript that addresses all the points raised by the two reviewers. 

We would appreciate receiving your revised manuscript by Jan 18 2020 11:59PM. To enhance the reproducibility of your results, we recommend that if applicable you deposit your laboratory protocols in protocols.io, where a protocol can be assigned its own identifier (DOI) such that it can be cited independently in the future. For instructions see: http://journals.plos.org/plosone/s/submission-guidelines#loc-laboratory-protocols

We look forward to receiving your revised manuscript.

Kind regards,

Jose Melo-Cristino, M.D., Ph.D.

Academic Editor

PLOS ONE

Journal Requirements:

3. In ethics statement in the manuscript and in the online submission form, please provide additional information about the patient records/samples used in your retrospective study. Specifically, please ensure that you have discussed whether all data/samples were fully anonymized before you accessed them and/or whether the IRB or ethics committee waived the requirement for informed consent. If patients provided informed written consent to have data/samples from their medical records used in research, please include this information.

4. Your ethics statement must appear in the Methods section of your manuscript. If your ethics statement is written in any section besides the Methods, please move it to the Methods section and delete it from any other section. Please also ensure that your ethics statement is included in your manuscript, as the ethics section of your online submission will not be published alongside your manuscript.

5. Please amend your manuscript to include your abstract after the title page.

6. Thank you for stating the following in the Competing Interests section:

This work is an investigator-initiated study that was supported by grants from: - The Landspítali University Hospital Research Fund (MÁH) https://www.landspitali.is

 - GlaxoSmithKline Biologicals SA (KGK, HE, ÁH), https://www.gsk.com

We note that you received funding from a commercial source: GlaxoSmithKline Biologicals SA

Reviewers' comments:

Reviewer's Responses to Questions

**Comments to the Author**

1. Is the manuscript technically sound, and do the data support the conclusions?

Reviewer #1: Yes

Reviewer #2: Yes

2. Has the statistical analysis been performed appropriately and rigorously? 

Reviewer #1: Yes

Reviewer #2: Yes

3. Have the authors made all data underlying the findings in their manuscript fully available?

Reviewer #1: Yes

Reviewer #2: Yes

4. Is the manuscript presented in an intelligible fashion and written in standard English?

Reviewer #1: Yes

Reviewer #2: Yes

5. Review Comments to the Author

Reviewer #1: The authors presented interesting data, regarding the epidemiology of pneumococcal infections and the evolution of antimicrobial resistance, after the introduction of a conjugate vaccine. However, here seem to be some aspects that need further clarification:

1. Materials and Methods: the authors should explain the method of identification of S. pneumoniae in cultures.

2. Materials and Methods, line 89: “serotyping was done using agglutination with Pneumotest-Latex and/or latex antisera for specified serotypes or pools and/or PCR” – why? Which serotypes were detected by each of the methods?

3. Materials and Methods, line 97: isolates defined as non-encapsulated when positive for lytA and negative for capsular genes. There are other streptococcal species carrying the lytA gene. Did the authors confirmed by any other method the identification of these isolates? Furthermore, non-encapsulated isolates ranked third among NVT(n=34), so this should be clarified.

4. Results, line 153: the number of PNSP changed from 41.1% in 2011 to 20.8% in 2014, but then increased again to 33.1% in 2017. Is the whole period, is really there a decreasing trend? As this is the main message of the paper, I believe it should be further discussed. Furthermore, the dynamics of the serotypes can add extra discussion to this, since the 19F isolates, the main serotype contributing to resistance, decreased form 2011 to 2016, but then increased again in 2017.

5. Analysis of figure 1 seems to suggest that vaccination was more effective in decreasing the proportion of middle ear infections than that of LRT infections. Can the authors please comment on this?

Reviewer #2: The authors describe changes in PNSP in Iceland from 2011-2017 post PCV-10 introduction. Overall the paper is well written, and results described and discussed is in a clear, concise manner.

Specific comments

Abstract. Last paragraph. “linage” should be “lineage”

Line 48-57. The paper describes changes in PNSP as percentages of cases throughout the paper. You include population denominators in your methods section. Your population sizes are similar for 2011 and 2017 but showing annual incidence (number of cases/100,000 population) would be a better comparison for showing differences over time.

Line 72. Interesting that you included pleural fluid in the LRT group. Why not as sterile body fluid?

Line 74. Isolates from “sinuses”? You excluded nasopharynx and throat but kept in sinuses? Why?

Line 159. I assume >0.6-0.5 mg/L should be >0.06-0.5 mg/L?

Fig 3, Please correct your lower values from 0.6 to 0.06 in the legend as well as in the figure.

6. PLOS authors have the option to publish the peer review history of their article (what does this mean?). If published, this will include your full peer review and any attached files.

Reviewer #1: No

Reviewer #2: No

---

## [Author Response · Author response to Decision Letter 0]

16 Jan 2020

Response from the authors to the reviewers

Journal Requirements

Response

The manuscript has been formatted according to PLOS ONE´s style requirements.

The names of institutions have been amended according to PLOS requirements as follows:

Faculty of Medicine, University of Iceland, Reykjavík, Iceland

Department of Clinical Microbiology, Landspitali University Hospital, Reykjavík, Iceland

BioMedical Centre of the University of Iceland, Reykjavik, Iceland

Children´s Hospital Iceland, Landspitali University Hospital, Reykjavík, Iceland

Response

Captions for our supporting information have been included at the end of the manuscript as follows:

S1 Table. Annual numbers and proportions of samples, total number of pneumococcal isolates, penicillin non-susceptible pneumococci (PNSP) and there of vaccine serotypes (VT) and non-vaccine serotypes (NVT), all according to sampling site.

S2 Table. Annual numbers and proportions of samples, total number of pneumococcal isolates, penicillin non-susceptible pneumococci (PNSP) and there of vaccine serotypes (VT) and non-vaccine serotypes (NVT), all according to age groups.

S3 Table. The ten most common serotypes of penicillin non-susceptible pneumococci (PNSP) in 2011-2017 and non-encapsulated (NESp) PNSP. Other are the remaining 16 serotypes that were detected in ≥5 isolates each during the study period. All of these are non-vaccine serotypes except for one isolate of 9V in 2011. 

3. In ethics statement in the manuscript and in the online submission form, please provide additional information about the patient records/samples used in your retrospective study. Specifically, please ensure that you have discussed whether all data/samples were fully anonymized before you accessed them and/or whether the IRB or ethics committee waived the requirement for informed consent. If patients provided informed written consent to have data/samples from their medical records used in research, please include this information.

Response

Additional information about samples and ethics was included as follows:

The samples were bacterial isolates and all patient data was fully anonymized before being were analysed. The National Bioethics Committee waived the requirement for informed consent.

4. Your ethics statement must appear in the Methods section of your manuscript. If your ethics statement is written in any section besides the Methods, please move it to the Methods section and delete it from any other section. Please also ensure that your ethics statement is included in your manuscript, as the ethics section of your online submission will not be published alongside your manuscript.

Response

The ethics statement has been written as separate sub chapter in the Material section as follows:

Ethics statement

The study was approved by The National Bioethics Committee (VSNb2013010015/03.07) and the appropriate authorities at the Landspitali University Hospital, Iceland. The samples were bacterial isolates and all patient data was fully anonymized before being analysed. The National Bioethics Committee waived the requirement for informed consent.

5. Please amend your manuscript to include your abstract after the title page.

Response

The abstract has been included directly after the title page.

6. Thank you for stating the following in the Competing Interests section:

This work is an investigator-initiated study that was supported by grants from: - The Landspítali University Hospital Research Fund (MÁH) https://www.landspitali.is

 - GlaxoSmithKline Biologicals SA (KGK, HE, ÁH), https://www.gsk.com

We note that you received funding from a commercial source: GlaxoSmithKline Biologicals SA

Within this Competing Interests Statement, please confirm that this does not alter your adherence to all PLOS ONE policies on sharing data and materials by including the following statement: "This does not alter our adherence to PLOS ONE policies on sharing data and materials.” (as detailed online in our guide for authors

Response

The Competing Interest Section now reads as follows:

This work is an investigator-initiated study that was supported by grants from: - The Landspítali University Hospital Research Fund (MÁH) https://www.landspitali.is

 - GlaxoSmithKline Biologicals SA https://www.gsk.com

The authors declare that this does not alter our adherence to PLOS ONE policies on sharing data and materials. Neither GlaxoSmithKline Biologicals SA, nor Landspítali University Hospital Research Fund had any role in study design, data collection and analysis, decision to publish, or preparation of the manuscript. 

Response to reviewer #1

1. Materials and Methods: the authors should explain the method of identification of S. pneumoniae in cultures.

Response 

We agree and added text after the first sentence in the chapter Bacterial isolates and samples as follows (line 99 in manuscript with track changes):

The pneumococci were cultured and identified from routine patient specimens using conventional methods, i.e. plated on two 5% horse blood agar plates (Oxoid, Hampshare, UK), one incubated in a 5% CO2 and the other one anaerobically. Identification was done using optochin test (and bile solubility if unclear). All isolates from 2016-2017 were also identified with MALDI-Tof. 

2. Materials and Methods, line 89: “serotyping was done using agglutination with Pneumotest-Latex and/or latex antisera for specified serotypes or pools and/or PCR” – why? Which serotypes were detected by each of the methods?

Response

We agree that this needs clarification and amended Material and Methods as follows (line 122 in manuscript with track changes):

Oxacillin resistant isolates were routinely screened for vaccine serotypes using agglutination with Pneumotest-Latex and/or latex antisera for specified serotypes or pools (Statens Serum Institute, Copenhagen). Isolates that did not belong to vaccine serotypes were serotyped using PCR. 

3. 

Materials and Methods, line 97: isolates defined as non-encapsulated when positive for lytA and negative for capsular genes. There are other streptococcal species carrying the lytA gene. Did the authors confirm by any other method the identification of these isolates? Furthermore, non-encapsulated isolates ranked third among NVT(n=34), so this should be clarified.

Response

During the last two years of the study, MALDI-Tof was used in addition to optochin testing for species identification. In an earlier study of our group (Quirk 2019), a large proportion of the isolates were analysed using whole genome sequencing, confirming the identification and the absence of the capsule.

Following information was added to the results (line 269 in manuscript with track changes):

All the NEsp isolates originated from adults except one from a 12 years old child, 30 of the isolates were from the LRT and three from conjunctiva.

Following text was added to the discussions directly after discussion of NVTs (line 324 in manuscript with track changes):

Interestingly NESp were not found in children while they were among the most common pneumococci in adults. This is in contrast to findings of other investigators who have reported an increase in children after initiation of vaccination (Takeuchi 2019, Keller 2016). In an earlier study of our group a large proportion of the isolates were analysed using whole genome sequencing, confirming the identification and the absence of the capsule (Quirk 2019).

4. Results, line 153: the number of PNSP changed from 41.1% in 2011 to 20.8% in 2014, but then increased again to 33.1% in 2017. Is the whole period, is really there a decreasing trend? As this is the main message of the paper, I believe it should be further discussed. Furthermore, the dynamics of the serotypes can add extra discussion to this, since the 19F isolates, the main serotype contributing to resistance, decreased from 2011 to 2016, but then increased again in 2017.

Response

The increase in PNSP during the last years of the study are due to increase in NVTs, mainly due to an increase in prevalence of 6C and 15A. The dominating serotype in 2011 was 19F that constantly decreased to 13 isolates in 2016 and their number was 15 in 2017. Of the isolates in 2016 and 2017, two isolates originated from children each year all the other were from adults. We do not know the vaccine status of the children and no change in vaccination recommendations for adults was implemented where the serotype still lingered. We cannot speculate on this low increase of 19F in 2017 as their numbers are so low in 2016 and 2017, but we agree that this needs clarification.

Following sentences sentencewereadded in the discussion (line 290 in manuscript with track changes):

Isolates of serotype 19F were still the majority of PNSP of VT, although only a fraction of their number in 2011, all originating from adults except one. Isolates of other VTs in 2017 were one isolate each of serotype 6B and 14. 

5. Analysis of figure 1 seems to suggest that vaccination was more effective in decreasing the proportion of middle ear infections than that of LRT infections. Can the authors please comment on this?

Response

The difference can be explained by the difference in the age of the patients. Otitis media is much more common in children. Accordingly, most of the ME samples originate from children. On the other hand, it is difficult to obtain samples from the LRT of small children and pulmonary infection is relatively common in the oldest patients. Accordingly, most of the lower respiratory tract samples originate from adults. A herd effect from the vaccination of children can explain the reduction that is later seen in LRT isolates.

Reviewer #2

1. Abstract. Last paragraph. “linage” should be “lineage”

Response 

The authors thank for this comment and have made appropriate correction.

2. Line 48-57. The paper describes changes in PNSP as percentages of cases throughout the paper. You include population denominators in your methods section. Your population sizes are similar for 2011 and 2017 but showing annual incidence (number of cases/100,000 population) would be a better comparison for showing differences over time

Response

The authors agree and thank for this comment. We have added the number of isolates of PNSP/100,000 inhabitants and the number of isolates of VTs of PNSP/100,000 inhabitants to supplementary table 2 and added this information into the results as appropriate (lines 223-228 and 26-238 in manuscript with track changes):

3. Line 72. Interesting that you included pleural fluid in the LRT group. Why not as sterile body fluid?

Response

We agree that pleural fluid is often included in sterile body fluids, however we have considered these LRT samples in our other studies and decided to do the same here.

4. Line 74. Isolates from “sinuses”? You excluded nasopharynx and throat but kept in sinuses? Why?

Response 

The authors excluded samples from nasopharynx for two reasons. The first being that during the study period, recommendations to physicians to send nasopharyngeal swabs changed. In the first years no specific recommendations were set, while in the later years the recommendation to physicians were to only send nasopharyngeal swabs from patients with treatment failure after using recommended antimicrobial prescriptions, or from patients with previously known antimicrobial resistant pneumococcal infection. The second reason is that samples from nasopharynx do not distinguish between carrier state and infection. On the other hand, samples from sinuses are mainly taken by otorhinolaryngologists in suspected infection (sinusitis).

5. Line 159. I assume >0.6-0.5 mg/L should be >0.06-0.5 mg/L?

Fig 3, Please correct your lower values from 0.6 to 0.06 in the legend as well as in the figure.

Response

The authors thank for this comment and have made appropriate corrections.

---

## [Decision Letter · Decision Letter 1]

23 Jan 2020

PONE-D-19-30435R1

Reduction of antimicrobial resistant pneumococci seven years after introduction of pneumococcal vaccine in Iceland

PLOS ONE

Dear Professor Hjálmarsdóttir,

Thank you for submitting your manuscript to PLOS ONE. After careful consideration, we feel that it has merit but does not fully meet PLOS ONE’s publication criteria as it currently stands. Therefore, we invite you to submit a revised version of the manuscript that addresses the points raised during the review process.

We invite you to submit a revised version of the manuscript that addresses all the important points raised by the reviewer.

We would appreciate receiving your revised manuscript by Mar 08 2020 11:59PM. To enhance the reproducibility of your results, we recommend that if applicable you deposit your laboratory protocols in protocols.io, where a protocol can be assigned its own identifier (DOI) such that it can be cited independently in the future. For instructions see: http://journals.plos.org/plosone/s/submission-guidelines#loc-laboratory-protocols

We look forward to receiving your revised manuscript.

Kind regards,

Jose Melo-Cristino, M.D., Ph.D.

Academic Editor

PLOS ONE

Reviewers' comments:

Reviewer's Responses to Questions

**Comments to the Author**

1. If the authors have adequately addressed your comments raised in a previous round of review and you feel that this manuscript is now acceptable for publication, you may indicate that here to bypass the “Comments to the Author” section, enter your conflict of interest statement in the “Confidential to Editor” section, and submit your "Accept" recommendation.

Reviewer #1: (No Response)

2. Is the manuscript technically sound, and do the data support the conclusions?

Reviewer #1: Yes

3. Has the statistical analysis been performed appropriately and rigorously? 

Reviewer #1: Yes

4. Have the authors made all data underlying the findings in their manuscript fully available?

Reviewer #1: Yes

5. Is the manuscript presented in an intelligible fashion and written in standard English?

Reviewer #1: Yes

6. Review Comments to the Author

Reviewer #1: In the response to the reviewers, the authors agreed that it was important to discriminate which serotypes were detected by each of the methods. In the revised version, the authors stated that “Isolates that did not belong to vaccine serotypes were serotyped using

PCR. The PCR was done as sPCR, or mPCR including a panel of all the serotypes included in the PHiD-CV and selected serotypes previously detected in our studies” – this is contradictory and needs further clarification.

It is clear that the authors identified vaccine serotypes, but among the others, only some were detected, while others were simply classified as NVT. I believe it would be important to know which NVT can the authors identify by the methods used, since one of the main messages of the paper is “NVT in the last years of the study is a cause of concern”. For example, serotype 8 is a NVT emerging in adult IPD in some countries, and the authors never mentioned this serotype. Is it absent from the collection or is simply not included in the serotyping scheme? Additionally, serotypes 35B and 11A have been associated with PNSP, but seem to be also absent from your collection, but they could also be not included in the scheme. I believe this should be clarified, because it is important to know which NVT are increasing.

7. PLOS authors have the option to publish the peer review history of their article (what does this mean?). If published, this will include your full peer review and any attached files.

Reviewer #1: No

---

## [Author Response · Author response to Decision Letter 1]

26 Feb 2020

Response to reviewer #1

In the response to the reviewers, the authors agreed that it was important to discriminate which serotypes were detected by each of the methods. In the revised version, the authors stated that “Isolates that did not belong to vaccine serotypes were serotyped using PCR. The PCR was done as sPCR, or mPCR including a panel of all the serotypes included in the PHiD-CV and selected serotypes previously detected in our studies” – this is contradictory and needs further clarification.

It is clear that the authors identified vaccine serotypes, but among the others, only some were detected, while others were simply classified as NVT. I believe it would be important to know which NVT can the authors identify by the methods used, since one of the main messages of the paper is “NVT in the last years of the study is a cause of concern”. For example, serotype 8 is a NVT emerging in adult IPD in some countries, and the authors never mentioned this serotype. Is it absent from the collection or is simply not included in the serotyping scheme? Additionally, serotypes 35B and 11A have been associated with PNSP, but seem to be also absent from your collection, but they could also be not included in the scheme. I believe this should be clarified, because it is important to know which NVT are increasing.

Response 

The authors misunderstood the reviewer’s comment. We realise this now and agree that the article will benefit from including information on the results from serotyping of all PNSP isolates. For that purpose, we have amended Supplementary table 3 and it now includes these results. Furthermore, we have added a list of all serotypes included in the mPCR panel to the chapter material and methods in lines 124-127 of the manuscript as follows: 

The PCR was done as sPCR, or mPCR including a panel of all the serotypes included in the PHiD-CV and selected serotypes previously detected in our studies - serotypes 1, 3, 4, 5, 6A, 6B, 6C, 6D, 6E, 7F, 8, 9V, 9N, 10A, 10F, 11A, 12F, 14, 15A, 15B/C, 16F, 17F, 18A/B/C/F, 19A, 19B/C, 19F, 20A/B, 21, 22F, 23A, 23B, 23F, 24F, 29, 31, 33F,33B/D, 34,35B, 35F, 35(25F), 42 (35A/C), 47A [8, 9, 24-26].

Six isolates were positive in lytA, cpsA and cpsB PCRs but negative in the serotype specific PCRs, this is also clarified in the table. Four isolatets of serogroup 15 and two isolates positive to pool G were further serotyped using PCR. The isolates of serogroup 15 were of serotype 15A and the isolates belonging to pool G were of serotype 35B. This was amended in the table and in the manuscript where lines 265-267 now read as follows:

The second most common NVT was serotype 15A with 17/516 (3.3%) isolates, increasing with time. All the 15A isolates were multi-resistant and had similar antibiograms as the serotype 6C isolates. The third was serotype 19A with 15/516 (2.9%) isolates, 1-4 isolates per year over the study period.

Regarding the reviewer’s comment on serotype 8, it is included in the mPCR panel but no isolate of serotype 8 was detected. Only two isolates of serotype 11A were detected, but serotype 35B ranks number ten of PNSP serotypes.

The title of supplementary table 3 has been amended as follows:

Annual distribution of pneumococcal serotypes and non-encapsulated Streptococcus pneumonia (NESp) defined as penicillin non-susceptible pneumococci (PNSP) in 2011-2017, number of isolates (n) and proportions (%).

---

## [Editor Report · Decision Letter 2]

27 Feb 2020

Reduction of antimicrobial resistant pneumococci seven years after introduction of pneumococcal vaccine in Iceland

PONE-D-19-30435R2

Dear Dr. Hjálmarsdóttir,

We are pleased to inform you that your manuscript has been judged scientifically suitable for publication and will be formally accepted for publication once it complies with all outstanding technical requirements.

With kind regards,

Jose Melo-Cristino, M.D., Ph.D.

Academic Editor

PLOS ONE
---

## [Editor Report · Acceptance letter]

3 Mar 2020

PONE-D-19-30435R2 

Reduction of antimicrobial resistant pneumococci seven years after introduction of pneumococcal vaccine in Iceland 

Dear Dr. Hjálmarsdóttir:

I am pleased to inform you that your manuscript has been deemed suitable for publication in PLOS ONE. Congratulations! Your manuscript is now with our production department. 

With kind regards,

on behalf of

Prof. Jose Melo-Cristino 

Academic Editor

PLOS ONE